# Culture Media Composition Influences the Antibacterial Effect of Silver, Cupric, and Zinc Ions against *Pseudomonas aeruginosa*

**DOI:** 10.3390/biom12070963

**Published:** 2022-07-09

**Authors:** Justyna Rewak-Soroczynska, Agata Dorotkiewicz-Jach, Zuzanna Drulis-Kawa, Rafal J. Wiglusz

**Affiliations:** 1Institute of Low Temperature and Structure Research, Polish Academy of Science, Okolna 2, 50-422 Wroclaw, Poland; j.rewak@intibs.pl; 2Department of Pathogen Biology and Immunology, University of Wroclaw, Przybyszewskiego 63/77, 51-148 Wroclaw, Poland; zuzanna.drulis-kawa@uwr.edu.pl

**Keywords:** bacterial culture media, metal ions, antibacterial activity, *Pseudomonas aeruginosa*

## Abstract

Different metals, such as silver (Ag), copper (Cu), and zinc (Zn), have been broadly investigated as metals and cations used both in medicine and everyday life due to their broad spectrum of antibacterial activity. Although the antibacterial action of those metals and their ions is well known and studied, the main problem remains in the standardization of experimental procedures to determine the antimicrobial activity as bacteriological media composition might significantly influence the outcome. The presented study aimed to evaluate the appropriability of different culture media (four nutritionally rich and four minimal) in the testing of the antibacterial activity of Ag^+^, Cu^2+^, and Zn^2+^ ions against *Pseudomonas aeruginosa.* Our investigation revealed the influence of medium ingredients and the presence of phosphates, which significantly reduced the activity of tested metal ions. Moreover, the precipitate formation and decrease in pH in the minimal media were additionally observed. It was assumed that the most favorable medium for metal ion activity testing was Luria-Bertani complex medium and MOPS minimal medium.

## 1. Introduction

The ability to control bacterial infections is one of the recent century’s achievements. Unfortunately, the development of antibiotic resistance, especially in clinically important pathogens, has become a serious problem [1]. Amongst the most life-threatening pathogens, *Pseudomonas aeruginosa* arouses the greatest concern due to high intrinsic and acquired resistance supported by genome plasticity [2]. *P. aeruginosa* is a common pathogen in immunocompromised patients that causes serious nosocomial infections. Its pathogenic potential is mainly addressed to virulence determinants produced, and among them, adhesins, toxins, flagella, lipopolysaccharide (LPS), pili, and phospholipases are the most dangerous. Produced pigments are responsible for metal-scavenging (iron-scavenging pyoverdine) or reactive oxygen species production (pyocyanin) [3,4]. The virulence and biofilm production is under the control of complex signalling systems [5].

Nowadays, we are facing an urgent need for new treatment options against multidrug-resistant pathogens (MDR), such as *P. aeruginosa*. Among a variety of modern therapies considered [6], metals possess well-known antibacterial activity and the potential to act in synergy with other antibacterial agents. Additional application of the nanosized compounds is associated with their binding capability to a wide variety of cations and molecules, namely metal ions as well as pharmaceutical species, such as antibiotics, inflammatory drugs, anticancer and anti-metastatic drugs. This provides an opportunity to obtain a specialised delivery system with controlled release of therapeutics. Unfortunately, antibiotic-resistance development in the presence of specific metal ions has also been reported [7,8]. Nevertheless, in the era of omnipresent multidrug resistance, new strategies against bacterial infection are of the greatest importance [9]. The antibacterial activity of metals depends on various mechanisms, such as protein impairment, damage of cellular membrane, and disturbance in nutrient uptake or even genotoxicity. However, the most important and potent mechanism preventing the growth of microorganisms in the presence of metals is the interaction between the bacteria and free radicals generated [10].

Although there are a lot of ways that metals and their ions are developed as new and promising materials for different therapies, the antibacterial metal activity testing is still problematic. The culture media chosen for experiments are mainly selected basing on the nutritional requirement of tested bacteria. The majority of bacteriological media are usually a mixture of peptone, tryptone, yeast extract, meat extract, inorganic salts, and other undefined additives [11]. Moreover, demanding microorganisms should be cultivated on media enriched with blood or serum. Therefore, the common media used are nutritionally rich and have undefined chemical composition and, thus, can differ from one producer to another or even between the production batch [12]. Contrarily, the minimal media (synthetic) have a defined chemical composition with specific carbon and nitrogen source, phosphates, vitamins, and trace elements but their application is limited to prototrophic bacteria cultivation [13,14].

The protocols for antibiotics testing [15] do not contain any standards for metal ion testing; therefore, it is difficult to choose a proper methodology. The standard techniques for antibiotics (agar disc diffusion method or broth dilution method) can be applied but some modifications are needed, especially for metal ions and their complexes [16,17].

The presented study aimed to evaluate the appropriability of different culture media in the testing of the antibacterial activity of metal ions. The strongest incentive to undertake such research was the lack of the comprehensive guidelines concerning the subject. Moreover, the literature analysis revealed that the researchers use different media; hence, it is hard to compare the obtained results. Therefore, in this study, three metal cations of known antibacterial activity (silver (Ag^+^), cupric (Cu^2+^), and zinc (Zn^2+^)) were used to supplement complex media (undefined) routinely used in microbiology: Luria Bertani Broth, Mueller Hinton Broth (MHB), Brain Heart Infusion Broth (BHI), and Trypticase Soy Broth (TSB). Furthermore, to evaluate the crucial components that influence the metal ion interactions, four minimal media with defined composition were also included in the study: MOPS, Davis, M9, and M63.

## 2. Materials and Methods

### 2.1. Bacterial Strains

The experiments were performed using *P. aeruginosa* reference strains: PAO1 and ATCC 27853 (American Type Culture Collection) and four clinical isolates: 15/3, 82/3, 9/5, and 14/3 showing different susceptibility patterns to antibiotics (Appendix A) [18].

### 2.2. Sources of Metal Ions

A cation of metals in the pure solutions was obtained using: AgNO_3_ (POCH, Gliwice, Poland), Cu(NO_3_)_2_^−^ 3H_2_O (Acros Organics, Geel, Belgium) and Zn(NO_3_)_2_·6H_2_O (CHEMPUR, Piekary Slaskie, Poland) as starting substrates. Weighed amounts of ingredients were dissolved in distilled water and the concentration of ions was set to 10 mg/mL per cation.

### 2.3. Culture Media

For antimicrobial activity testing the following complex culture media were used: Luria Bertani Broth (LB) (BioShop, Burlington, ON, Canada), Mueller Hinton Broth (MHB) (with 2.741 mg Ca^2+^/L and 5.282 mg Mg^2+^/L) (Oxoid, UK), Trypticase Soy Broth (TSB) (Biomaxima, Lublin, Poland) and Brain Heart Infusion Broth (BHI) (Oxoid, Basingstoke, UK). The composition of tested media is presented in Figure 1. All media were prepared according to the producer guidelines as non-diluted or 1:1 diluted with sterilized distilled water as it was proposed elsewhere [19]. The dilution process was applied to decrease the number of media components that were able to interact with tested ions.

Moreover, four minimal (synthetic) media were chosen for the analyses. The components that were used to prepare MOPS, M9, M63 and Davis media are listed in Appendix A, and the content of particular components is compared in Figure 2. All media were prepared according to the previously reported guidelines and pH of all minimal media was set at 7.2–7.4 using 10 M NaOH solution.

### 2.4. Ag^+^, Cu^2+^ and Zn^2+^ MIC Determination

Minimum inhibitory concentrations (MICs) of tested metal ions were determined using a standard microdilution method (https://www.eucast.org/ast_of_bacteria/; accessed on 17 February 2020) in a 96-well plate with modifications [20]. All strains were incubated overnight on MHA plates (Biomaxima, Lublin, Poland) at 37 °C and suspended in saline to the OD_600_ = 0.1 (spectrophotometer HACH LANGE GmbH). Ag^+^, Cu^2+^ and Zn^2+^ stocks were diluted in a specific medium to obtain concentrations in a range between 10 and 1000 µg/mL (by 10 µg/mL each step). Moreover, lower concentrations of Ag^+^ were prepared (0.00625–10 µg/mL) regarding higher bacterial susceptibility observed. As such, 100 µL of each solution was added to the well and inoculated with 10 µL of bacteria to a final density of 1 × 10^6^ CFU/mL and incubated at 37 °C with agitation for 24 h. Negative controls (without bacteria) were prepared separately for each ion concentration in each medium and then their absorbance values were subtracted from the corresponding sample absorbance value. Pure media with bacteria were also prepared as growth controls. After 24 h of incubation, the optical density at 600 nm was measured (Varioskan LUX, Thermo Fisher Scientific, Waltham, MA, USA). MIC values were determined as the lowest concentration tested with the lack of turbidity (visually and OD_600_ measurement confirmed). All experiments were performed in triplicate on three different days.

### 2.5. Metal–Medium Interactions

Here, 0.2 mL of silver, cupric, or zinc nitrate solutions was mixed with 1.8 mL of particular media to a final volume of 2 mL in a 12-well plate to obtain a final ion concentration of 1 mg/mL. Visual observations of the colour changes as well as precipitation processes were made at the beginning and after 24 h of incubation with agitation (100 rpm) at 37 °C. Moreover, to evaluate how the minimal media components react with metal ions causing visible changes (colour changes and/or precipitation processes) the solutions of each medium component were prepared at a concentration corresponding to their final concentration in the media. Results were verified after 24 h of incubation at 37 °C with agitation.

Moreover, the pH of tested samples, as well as the control sample (media without metal ions added), was measured. For LB and MOPS media a wider range of ion concentrations (10–1000 µg/mL) were prepared to examine dose-dependent pH changes. Distilled water was used instead of the medium as a control.

### 2.6. Bacterial Growth in Selected Media with a Modified pH Range

All strains were cultured overnight on MHA plates (Biomaxima, Lublin, Poland) at 37 °C and suspended in saline (0.9% NaCl, Chempur) to OD_600_ = 0.1 (spectrophotometer HACH LANGE GmbH, Berlin, Germany). Two media were selected for this experiment: minimal MOPS medium and complex LB medium. The media pH was set in a range between 5.0 and 8.0 with 0.5 steps using 36 % HCl and 10 M NaOH solutions (Mettler Toledo Quattro pH meter, MP220 Basic, Columbus, OH, USA). Further, 100 µL of the medium at a specific pH was added to the well and inoculated with 10 µL of bacterial solutions to a final density of 1 × 10^6^ CFU/mL. All experiments were performed in triplicate on three different days. The plates we incubated with agitation for 24 h at 37 °C and the absorbances (OD_600_) were measured (VarioSkan LUX, Thermo Fisher Scientific, Waltham, MA, USA). The statistical analysis was performed using one-way ANOVA and the Levene test, followed by the Tukey test in the OriginPro 2019b (OriginLab Corporation, Northampton, USA) software (*p* < 0.05).

### 2.7. The Correlation between Medium Composition and MIC Values of Tested Ions

The correlation between the number of media components (total content, organic content, phosphate content and chloride content) and MIC values against *P. aeruginosa* PAO1 and ATCC 27853 strains was also described in scatter graphs with linear fit and calculated R^2^ values (OriginPro 2018, OriginLab). The R^2^ values were interpreted as follows: R^2^ > 80%—strong correlation; 40–80%—moderate correlation; <40%—weak correlation or no correlation. The ranges were set according to the information found elsewhere [21,22].

## 3. Results

### 3.1. The Type of Culture Medium Influences the Antibacterial Activity of Metal Ions

The first part of this work was focused on the comparison of the antimicrobial efficacy of Ag^+^, Cu^2+^, and Zn^2+^ in various media, both complex and minimal. Additionally, the dilution of the complex media was applied to test whether the limitation in components amount per ml will influence the antibacterial activity of tested metals and whether that effect is equal in all media. Minimum inhibitory concentrations of tested metal ions against *P. aeruginosa* PAO1 and ATCC 27853 in tested media are listed in Table 1.

The best antimicrobial activity was observed for Ag^+^, regardless of the medium type, and among complex non-diluted media, the MICs ranged from 1.25 to 10 for LB and BHI media, respectively. Dilution of the media influenced the Ag^+^ activity leading to an almost two-fold MIC decrease, observed for all media except BHI. Similar observations were made for Cu^2+^ and Zn^2+^. MICs for Cu^2+^ ranged from 240 to 740 for LB and BHI, respectively, and from 110 to 370 in diluted medium. For Zn^2+^, the lowest MIC was detected in LB medium for *P. aeruginosa* ATCC 27853; however, in TSB and BHI, the growth inhibition was not detected, even at the highest concentration (1 mg/mL). In the diluted version, the MICs ranged from 150 to 720 for LB and BHI, respectively. MICs for Cu^2+^ and Zn^2+^ in LB and MHB were similar and relatively low but in TSB and BHI, were 2–3-times higher. For Zn^2+^ in non-diluted TSB and BHI media, MICs were outside of the scale at tested concentrations (>1000 µg/mL). Moreover, it can be noted that in 1:1 diluted media, the MIC values were approximately two-fold lower than in non-diluted analogues and it can be seen for all tested ions.

The analysis of media composition (Figure 1 revealed major differences in the content of organic and inorganic compounds between complex and minimal media. Among tested complex media, BHI and TSB are the most abundant in nutrients, especially in organic fraction, whilst the least rich were LB and MHB. It should be noted that the organic fraction in BHI was almost two-times higher than in LB (29.5 and 15 g/L, respectively). The total amount of medium ingredients is important; however, more informative is the analysis of medium quality, especially protein-based components and phosphate-based inorganic salts that differ between tested media. Therefore, in the least-nutritious medium among complexes (LB) (see Figure 1), the MICs were the lowest, and in the most abundant medium (BHI), the MICs were the highest, which clearly indicates that the cation activity may be influenced both by organic and inorganic components in the media.

For Ag^+^ and Cu^2+^, MICs were relatively lower in minimal media than in complex media. The MICs for Ag^+^ were similar for all tested minimal media and the values were very low (the lowest detected MIC was 0.00625 µg/mL per cation). Moderate differences were assessed for Cu^2+^ and the lowest MIC was evaluated in M63 medium (5 µg/mL) and the highest in Davis medium (100 µg/mL). Antimicrobial activity of Zn^2+^ could only be noted in minimal MOPS medium (160–170 µg/mL depending on the strain). In M9, M63 and Davis, Zn^2+^ gave heavy precipitation and bacterial turbidity could not be measured. Therefore, it was decided to investigate what interactions lie behind this phenomenon. The analysis of minimal media composition (Figure 2) revealed that Davis medium contains the most ingredients; moreover, the content of phosphate-based salts was also the highest among tested minimal media. Despite the highest variety of different components in the MOPS medium, the general amount of nutrients was the lowest among minimal media. All synthetic media contain phosphate-based salts since it is mandatory for bacterial growth; however, the contents in MOPS medium was also the lowest. In minimal media, it is necessary to provide proper pH balance and in most of them, it was obtained by phosphate’s presence; however, in MOPS medium, other components (MOPS and tricine) were used.

To verify the differences observed between obtained MICs for reference strains, four clinical isolates of *P. aeruginosa * (15/3, 82/3, 9/5, 14/3), having different susceptibility patterns to common antibiotics (see Appendix A), were selected. Based on preliminary results, complex LB and minimal MOPS media were chosen, as the MIC values for reference strains were the lowest and the analysis of their composition revealed that they are the least abundant in components that may potentially interact with metal cations. *P. aeruginosa* 15/3 and 82/3 were both antibiotic susceptible, contrary to 9/5 and 14/3, which were resistant to at least three groups of antibiotics (MDR—multidrug resistance) (see Appendix A) [23]. The MICs for tested metal ions are presented in Table 2 and the results support the tendency observed for the reference strains (Table 1).

### 3.2. The Chemical Interactions between Metal Ions and Culture Media Components

Precipitation observed during MIC measurement forced us to investigate whether metal ions interact with media components. As this part is crucial for a general understating of the process, it was decided to investigate all eight media instead of the previously chosen LB and MOPS only. The visible interactions between various media and Ag^+^, Cu^2+^, and Zn^2+^, seen as colour changes or precipitate presence, are presented in Figure 3.

In complex media, an increase in turbidity was noted, but only after overnight agitation and the most visible changes were observed for Cu^2+^ (brown spots visible at the bottom of the wells). On the contrary, in minimal media, the changes occurring after supplementation of the medium with a metal ion (Ag^+^, Cu^2+^, Zn^2+^) were more visible than in complex ones. These changes mainly involved precipitation and turbidity increase and were ion dependent. The most colourful and solid precipitates were formed by Ag^+^, whilst those formed by Cu^2+^ (blueish) and Zn^2+^ (white) were more scattered.

To explain observed changes, the particular media components were mixed with 1 mg/mL of Ag^+^, Cu^2+^ and Zn^2+^ ions and the observed precipitation is described in Appendix A. Moreover, the theoretical analysis of interactions between particular media components and metal ions was performed and the chemical reactions are listed in Table 3.

Changes in the minimal media (MOPS and M9) were observed because of Ag^+^ and Cl^−^ ion precipitation to white, curdy silver chloride (AgCl), formed in neutral and acid solutions. In Davis and M63 media, yellow-brown precipitates, probably of silver phosphate, were observed. In the case of Cu^2+^ addition, the observed precipitation in MOPS medium is related to iron (II) and iron (III) phosphates. Additionally, cupric(I) oxides in all studied minimal media were observed. It is related to D-glucose that reduces Cu^2+^ to Cu^+^ and precipitates in neutral copper solutions as brown-red copper(I) oxide. Moreover, Cu^2+^ ions form an aqua complex [Cu(H_2_O)_4_]^+^ and copper (II) citrate Cu(HCit)_n_^n(d−3)+2^. In the case of Zn^2+^, phosphate solution precipitates into gelatinous, tertiary zinc phosphate in all evaluated media that soon becomes crystalline (Table 3).

The pH changes may also indicate the chemical reaction taking place; thus, the pH of all tested media was measured at 1 mg/mL concentration of metal ions (Table 4). As can be seen, no significant pH differences were observed for minimal media in all cases. The same situation was observed for diluted and non-diluted complex media for Ag^+^, whereas for Cu^2+^ and Zn^2+^, the pH value dropped down with the lowest values around pH 3.5 for Cu^2+^ in LB.

For further consideration about buffering properties of tested media, previously chosen LB and minimal MOPS media were selected. Different concentrations of ions were added and the pH range was measured. Deionized water was taken as a control (Figure 4). A similar pattern of pH decrease was observed for both water and LB medium. Therefore, it can be suspected that the LB medium had low buffering properties for tested ions. In the MOPS medium, pH changes occurring with the increase in the metal ion concentration were not as visible as for water and LB. In all tested media, the lowest pH (more acidic) was obtained for Cu^2+^. A moderate pH decrease was observed for Zn^2+^ ions. Interestingly, Ag^+^ ions did not influence the pH value of tested culture media; however, the pH decrease was detected after dissolving the AgNO_3_ stock in water (control sample).

### 3.3. Bacterial Growth in the Modified pH Range Depends on Media Composition

Considering pH changes that occur in the culture media supplemented with metal ions (especially Cu^2+^), the evaluation of bacterial growth in the media with different pH was examined. The ability of *P. aeruginosa* PAO1 and ATCC 27853 strains to grow in LB and MOPS minimal medium with different ranges of pH were determined to eliminate the effect of low pH on growth inhibition. As observed previously, higher concentrations of ions determined lower pH values of media. Our concern was to prove that the antibacterial effect was obtained as the result of ion activity, not the pH variation (Figure 5).

In LB medium, no differences were observed regarding the pH values tested, which suggests that LB is an appropriate medium for microbial cultivation with metal ions, which can cause medium acidification (such as Cu^2+^). The pH lowering in the range of media with tested ions did not affect the growth of bacteria; thus, the obtained results confirmed the assumption that the antibacterial effect results from metal ion presence, regardless of pH changes that they may trigger. A different observation was made for minimal MOPS medium where the highest absorbance was measured at a pH range between 6.5 and 7.5 for both tested strains and their growth was significantly reduced at the pH below 6.5. Further studies should be undertaken to explain that effect.

### 3.4. Correlation between Medium Composition and MIC Values of Tested Ions

The above analyses show that the antibacterial activity of tested metal ions is strictly dependent on the medium in which they are suspended. Additionally, it has also been shown that interactions between the media (and their single components) and cations (Ag^+^, Cu^2+^ and Zn^2+^ ions) were detectable as colour change, increased turbidity, precipitation or pH change. In order to summarize the observations and draw conclusions, it was decided to correlate the content of components in all tested media (minimal, complex and 1:1 diluted complex) with the corresponding MIC values for Ag^+^ and Cu^2+^. The Zn^2+^ was excluded from the analysis due to undetermined MIC values in all tested media (>1000 µg/mL). Based on the observations made in previous experiments (especially precipitation and turbidity occurrence), it was decided to compare the different content of phosphates, chlorides, organic components and the total content of components in the media (g/L), which in the complex media is not known; therefore, the analysis only shows the content of phosphate salts and NaCl declared on the label. The obtained R^2^ values (Appendix A) suggest a positive correlation between the total amount of media components and MIC values for Ag^+^ and Cu^2+^ ions; however, divalent ions seem to be more influenced by the medium (R^2^ = 0.89—strong correlation) than monovalent ones (R^2^ = 0.58), for which the correlation is moderate. The correlation is quite similar for organic content and phosphates as well (0.82 and 0.86 for Cu^2+^ and 0.55 and 0.59 for Ag^+^ ions). A significantly weaker correlation was observed between the chloride content and MICs, especially for Ag^+^ (0.32). For Cu^2+^ ions, the obtained R^2^ value (0.61) could be interpreted as moderate, but it should be taken into account that it is much lower than the correlation calculated for other groups of components.

## 4. Discussion

The presented study aimed to evaluate the applicability of different complex and minimal culture media for antibacterial activity of Ag^+^, Cu^2+^ and Zn^2+^ against *P. aeruginosa*. This work was motivated by the lack of precise protocols describing the metal ion activity testing, especially in regard to the culture media used. Among complex media, LB, MHB, TSB and BHI were tested, and to evaluate minimal media utility, MOPS, M9, M62 and Davies were chosen.

The study revealed that antibacterial activity of Ag^+^, Cu^2+^ and Zn^2+^ varied depending on media components. Ag^+^ was the most active ion in both complex and minimal media, whilst Cu^2+^ presented a moderate antibacterial effect. The antibacterial activity of Zn^2+^ was the lowest both in complex and minimal media. Generally, MIC values for all tested metal ions were noticeably lower in minimal media than in complex ones, which follows the general knowledge [19,24]. Complex LB and minimal MOPS media supplemented with tested ions exhibited the lowest MIC values and, therefore, were chosen for further investigation against *P. aeruginosa* clinical strains. Unfortunately, even though low MIC values were also obtained in M9 and M63 for Cu^2+^ and Ag^+^, these media, as well as Davis, should not be used for metal ion activity testing, due to a possible precipitation effect, as observed for Zn^2+^ supplementation. The effect of precipitation was not observed in the case of MOPS medium. Hence, among minimal media, MOPS is the only reasonable choice.

The literature describes the use of both complex and minimal media in metal ion activity testing. Among complex media, the most commonly used are: MHB [25,26,27], LB [28,29], nutrient broth [30,31,32,33] and TSB [34]. In the work by Kaneko et al. (2007), apart from complex TSB, a minimal medium (basal medium 2, BM2) was also used [34].

The idea of diluting culture media to reduce the amount of components that may interact with metal cations is not new [19,34]. However, in this work, it was proven that the quantity of medium ingredients is far less important than the quality of constituents. It could be deduced that highly nutritious, mostly protein-based organic compounds that are found in TSB and BHI could interact with metal ions, reducing their antibacterial activity. Such a hypothesis was confirmed, to some extent, by diluting the media with water. Hence, even in diluted BHI, where the general amount of component was lower than in non-diluted LB, MIC values were high. Naturally, the dilution also reduces bacterial growth because of the nutrient limitation (data not shown); hence, this process should rather not be applied as laboratory standard procedure. Apart from peptide-based compounds, others, such as glucose, are also reported as medium components responsible for reducing the bioavailability of mainly di- or trivalent ions [14,35].

Using minimal media in metal ion testing could also be considered; however, it should be noted that not all bacterial species are capable of living in such conditions, which is a serious limitation. The quality and quantity of medium components should also be analysed before planning the experiment because, as was proven in this work, some components (mostly phosphate-based salts) interact with metal cations to such a great extent that they become useless as antimicrobials. Ag^+^ and Cu^2+^ are generally more active than Zn^2+^, so their MIC values were detected in all tested minimal media. Contrary to Ag^+^ and Cu^2+^, Zn^2+^ was effective only in less interactive MOPS medium. That difference may also lay in a stronger Zn^2+^:PO_4_^3−^ affinity (especially under specific conditions) but it needs to be confirmed in further study.

Examining the exact impact of complex media on metal ions is difficult, due to their undefined composition; hence, this work describes the interactions of Ag^+^, Cu^2+^ and Zn^2+^ only with minimal media components. The observed changes, such as precipitation, colour change, turbidity growth and pH change, may be the result of chemical interactions of particular medium components with tested ions. However, single-component analysis may clarify the medium–metal ion interactions. Hence, the turbidity growth, observed after supplementing MOPS and M9 medium with Ag^+^, was a result of AgCl white precipitate formation. In media with high phosphate (KH_2_PO_4_ and K_2_HPO_4_) content (M63 and Davis), yellow-silver phosphates, blueish cupric phosphates and white zinc phosphates were formed after supplementing the media with the metal ion solutions. A similar observation was previously made by Li et al. (2011) who investigated Zn^2+^ behaviour in phosphate-rich minimal media [35]. However, because Ag^+^ is characterised by very high antimicrobial activity, these limitations seem to be less important than in the case of Cu^2+^ and Zn^2+^. This statement is supported by MIC values that are almost the same for Ag^+^, despite the minimal medium used, whilst they vary for Cu^2+^ and Zn^2+^. Since phosphate-based components, which are often used as buffering agents, are highly interactive, they should be replaced with phosphate-free substitutes, such as MOPS, MES, HEPES, PIPES, Tris or Tricine. These substances, belonging to the family of Good’s buffers, are not likely to precipitate in the presence of free metal cations [14,36].

The investigations focused on metal ion behaviour in culture media were conducted before; however, they do not provide all-encompassing information [14,19,20]. Many works are based on di- or trivalent ions; however, as proved in this work, monovalent ions, such as Ag^+^, also tend to react with medium components. Hasman et al. (2009), apart from describing the influence of the culture medium type on the metal toxicity level, also pointed out the relationship between metal ion activity and the medium pH. The pH value can, for example, influence the Cu^2+^ binding to medium components [19]. Thus, only experiments conducted in the same conditions (medium, pH, preparation procedure) may be compared to avoid drawing false conclusions. The analysis of the correlation between the content of components in the media and the MIC values confirmed that the activity of metal ions is strictly dependent on the selected medium. According to an available report [36], it might be assumed that divalent ions could be much more prone to changes in antibacterial activity than monovalent ones. Among the analysed groups of components, the highest correlation was observed for phosphates and organic components and the values are similar to those calculated for the total content of components (R^2^ = 0.58 for Ag^+^ and 0.88 for Cu^2+^). The effect of chlorides on the MIC values of Ag^+^ and Cu^2+^ was much weaker, especially in the case of Ag^+^, which is somewhat surprising given the fact that silver ions precipitate under the influence of Cl^−^. This may indicate the fact that the precipitation or the formation of complexes does not in itself cause a complete loss of the cation’s biological activity; however, such considerations go far beyond the subject of this paper. In the future, an important supplement to this research could be to study the amount of free metal cations that have the ability to affect the bacterial cell in specific media. Similar studies have already been carried out by Rathnayake et al. (2013), who showed that the amount of free Cu^2+^ in the TSB medium is very low, even below the detection level, compared to the minimal media based on Tris and MES buffers [14]. Moreover, it has been previously described that the amount of free Cu^2+^ (measured by ion-selective electrode) in the media could be even 10^7^–10^8^-times lower than total cupric concentration [19]. Total metal concentration in the medium could be measured by ICP OES (inductively coupled plasma optical emission spectroscopy) or AAS (atomic absorption spectrometry) but using these methods is not appropriate, taking into account the fact that bacteria are mainly affected by free metal ions Hence, the measurement of total metal concentration in the medium (AAS or ICP methods) should be followed by the analysis of the concentration of free metal cations measured by an ion-selective electrode (ISE). The phenomena observed in our work supplement the previously conducted research; however, there is still much work to be done before the question raised in this work can be fully answered.

## 5. Conclusions

Based on the results obtained in this work and the analysis in the literature, it can be concluded that minimal media with the lowest possible phosphate level (only to fulfil bacterial requirements) seem to be the best choice for metal ion antibacterial testing. Nonetheless, not all bacteria are capable of surviving in such nutritionally limited conditions and usage of a complex medium may be necessary, but only the simplest compositions, such as LB or MHB, could be recommended in this respect.

## Figures and Tables

**Figure 1 biomolecules-12-00963-f001:**
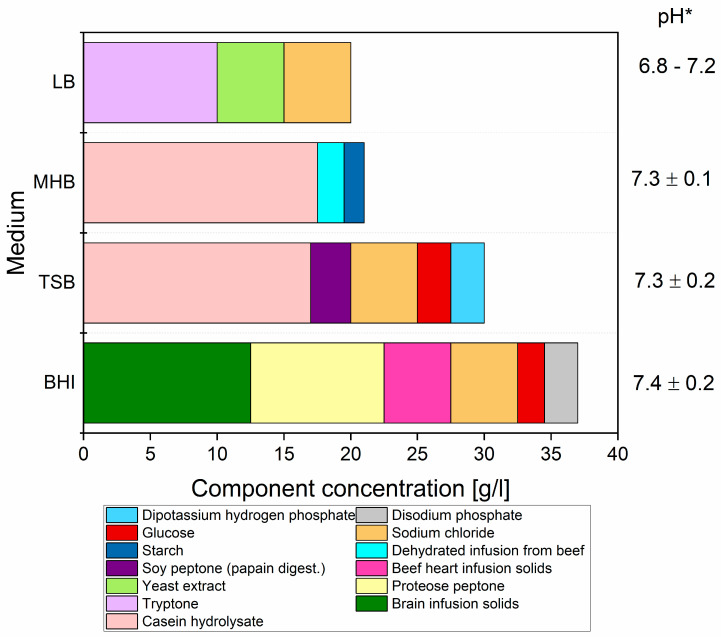
Concentration (g/L) of components in complexed media used for antibacterial metal activity testing. *—pH according to manufacturer.

**Figure 2 biomolecules-12-00963-f002:**
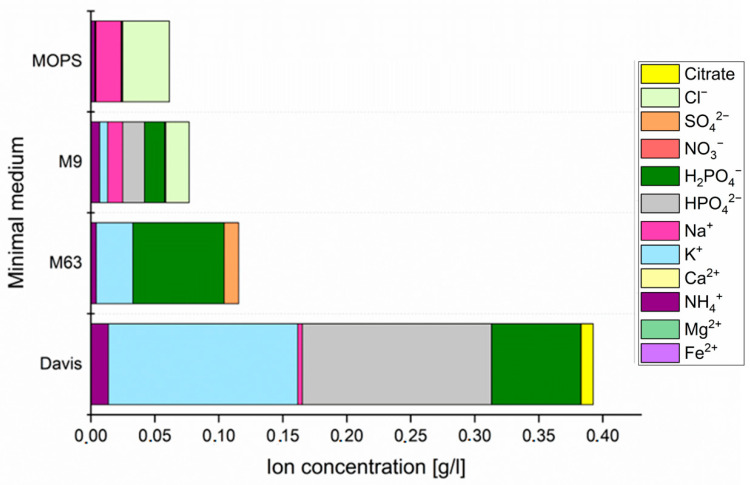
Concentration (g/L) of ions in minimal media used for antibacterial metal activity testing.

**Figure 3 biomolecules-12-00963-f003:**
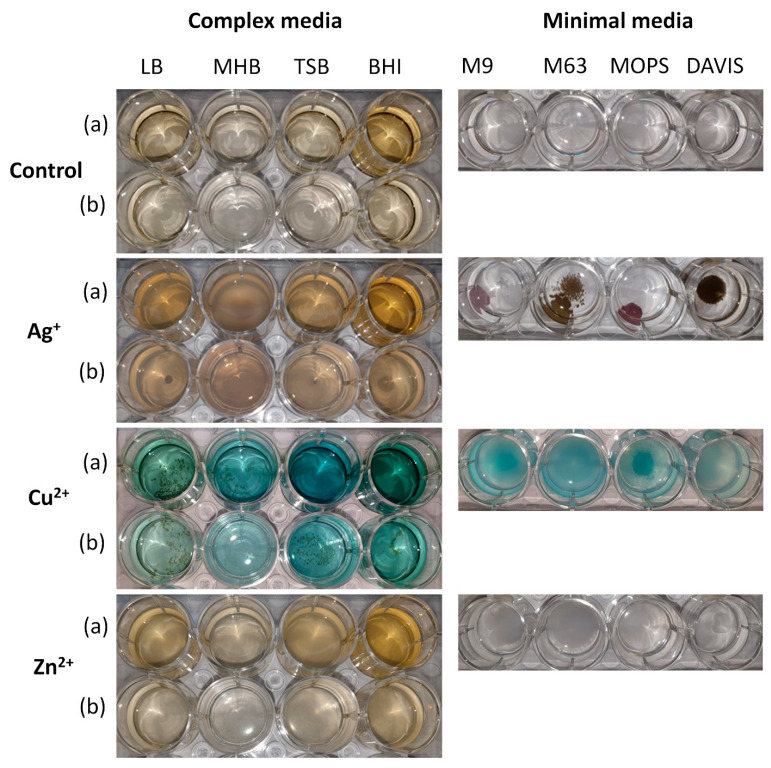
Colour changes and precipitates observed in different media (complex and minimal) mixed with Ag^+^, Cu^2+^, and Zn^2+^: (**a**) non-diluted medium, (**b**) 1:1 diluted with water.

**Figure 4 biomolecules-12-00963-f004:**
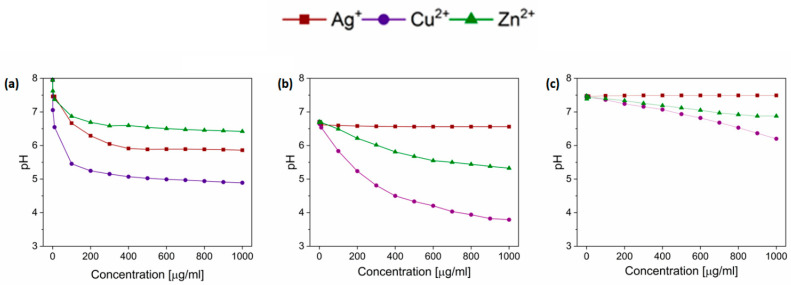
pH changes after addition of metal ions to chosen solutions: (**a**) distilled water, (**b**) non-diluted LB, (**c**) MOPS medium.

**Figure 5 biomolecules-12-00963-f005:**
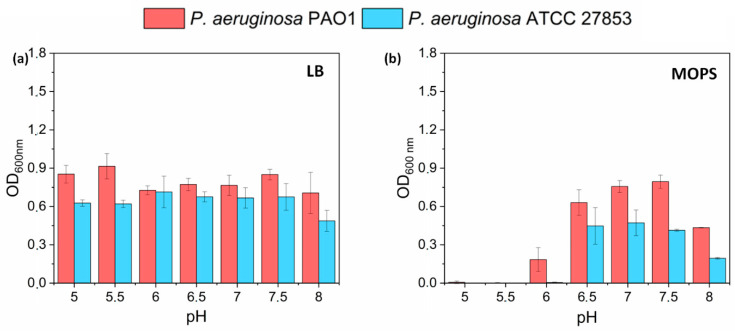
pH-dependent growth of *P. aeruginosa* PAO1 and ATCC 27853 strains in (**a**), LB (**b**), MOPS medium (mean ± sd, *n* = 3; for statistical analysis see Appendix A).

**Table 1 biomolecules-12-00963-t001:** Comparison of Ag^+^, Cu^2+^ and Zn^2+^ MICs in tested complex and minimal media.

Minimum Inhibitory Concentrations [µg/mL]				
Strain	Complex Media (Non-Diluted/1:1 Diluted)			
	LB	MHB	TSB	BHI
Ag^+^				
PAO1	1.25/0.625	2.5/0.625	5/1.25	10/10
ATCC 27853	1.25/1.25	2.5/1.25	5/2.5	10/10
Cu^2+^				
PAO1	240/110	290/130	690/340	730/360
ATCC 27853	250/120	290/140	690/340	740/370
Zn^2+^				
PAO1	350/160	370/160	>1000/550	>1000/720
ATCC 27853	250/150	330/160	>1000/490	>1000/700
	Minimal media			
	MOPS medium	M9 medium	M63 medium	Davis medium
Ag^+^				
PAO1	0.0125	0.0125	0.00625	0.0125
ATCC 27853	0.0125	0.025	0.0125	0.0125
Cu^2+^				
PAO1	50	20	5	100
ATCC 27853	70	20	20	100
Zn^2+^				
PAO1	170	>1000	>1000	>1000
ATCC 27853	160	>1000	>1000	>1000

**Table 2 biomolecules-12-00963-t002:** MIC of Ag^+^, Cu^2+^, Zn^2+^ ions in chosen media against *P. aeruginosa* clinical isolates.

MIC [µg/mL]		
Strain	LB (Non-Diluted/1:1 Diluted)	MOPS Medium
Ag^+^		
9/5	1.25/0.625	0.0125
14/3	1.25/0.625	0.025
82/3	1.25/0.625	0.0125
15/3	1.25/0.625	0.0125
Cu^2+^		
9/5	170/100	80
14/3	150/90	100
82/3	130/70	50
15/3	140/70	50
Zn^2+^		
9/5	280/150	180
14/3	250/100	120
82/3	250/110	190
15/3	250/110	170

**Table 3 biomolecules-12-00963-t003:** The chemical reactions of metal ions at a concentration with minimal media components.

**No.**	**MOPS Medium**	**Davis Medium**	**M9 Medium**	**M63 Medium**
Ag^+^				
1.	Ag^+^ + Cl^−^ → AgCl↓ (white)AgCl↓ + hν → Ag^0^ + Cl_2_(silver-grey)	3Ag^+^ + PO_4_^2−^ → Ag_3_PO_4_↓ (yellow-brown)	Ag^+^ + Cl^−^ → AgCl↓ (white)AgCl↓ + hν → Ag^0^ + Cl_2_ (silver-grey)	3Ag^+^ + PO_4_^2−^ → Ag_3_PO_4_↓ (yellow-brown)
2.	AgCl↓ + 2NH_3_·H_2_O → [Ag(NH_3_)_2_]^+^ + Cl^−^ + H_2_O (colourless)		AgCl↓ + 2NH_3_·H_2_O → [Ag(NH_3_)_2_]^+^ + Cl^−^ + H_2_O (colourless)	
3.	[Ag(NH_3_)_2_]^+^ + Cl^−^ + 2H^+^ → AgCl↓ + 2NH_4_^−^ (white)AgCl↓ + *hν* → Ag^0^ + Cl_2_ (silver-grey)		[Ag(NH_3_)_2_]^+^ + Cl^−^ + 2H^+^ → AgCl↓ + 2NH_4_^−^ (white)AgCl↓ + hν → Ag^0^ + Cl_2_ (silver-grey)	
4.	3Ag^+^ + PO_4_^2−^ → Ag_3_PO_4_↓ (yellow-brown)		3Ag^+^ + PO_4_^2−^ → Ag_3_PO_4_↓ (yellow-brown)	
Cu^2+^				
1.	Cu^2+^ + H_2_O → [Cu(H_2_O)_4_]^+^ (blue)	Cu^2+^ + n (H_2_Cit^−^) → Cu(HCit)_n_^n(d−3)+2^ + n (2-d)H^+^	Cu^2+^ + OH^−^ + C_6_H_12_O_6_ → Cu_2_O↓ + C_6_H_12_O_7_ + H_2_O (brown-red)	Cu^2+^ + OH^−^ + C_6_H_12_O_6_ → Cu_2_O↓ + C_6_H_12_O_7_ + H_2_O (brown-red)
2	4Fe^2+^ + 3HPO_4_^2−^ → FeHPO_4_↓ + Fe_3_(PO_4_)_2_↓ + H^+^ (white)	Cu^2+^ + OH^−^ + C_6_H_12_O_6_ → Cu_2_O↓ + C_6_H_12_O_7_ + H_2_O (brown-red)		
3	Fe^2+^ + NO_3_^−^ + H^+^ → Fe^3+^ + NO↑ + H_2_O (yellow)			
4	Fe^3+^ + HPO_4_^2−^ → FePO_4_↓ + H^+^ (yellow)			
5	Cu^2+^ + OH^−^ + C_6_H_12_O_6_ → Cu_2_O↓ + C_6_H_12_O_7_ + H_2_O (brown-red)			
Zn^2+^				
1.	3Zn^2+^ + 2HPO_4_^2−^ → Zn_3_(PO_4_)_2_↓ + 2H^+^ (white)	3Zn^2+^ + 2HPO_4_^2−^ → Zn_3_(PO_4_)_2_↓ + 2H^+^ (white)	3Zn^2+^ + 2HPO_4_^2−^ → Zn_3_(PO_4_)_2_↓ + 2H^+^ (white)	3Zn^2+^ + 2HPO_4_^2−^ → Zn_3_(PO_4_)_2_↓ + 2H^+^ (white)
2.		3Zn^2+^ + 2H_2_PO_4_^2−^ → Zn_3_(PO_4_)_2_↓ + 4H^+^ (white)	3Zn^2+^ + 2H_2_PO_4_^2−^ → Zn_3_(PO_4_)_2_↓ + 4H^+^ (white)	

**Table 4 biomolecules-12-00963-t004:** pH values of tested media supplemented with 1 mg/mL Ag^+^/Cu^2+^/Zn^2+^ nitrates.

	Control	Ag^+^	Cu^2+^	Zn^2+^
Complex media (non-diluted/1:1 diluted)				
LB	6.71/6.81	7.17/6.92	3.77/3.52	5.14/5.12
MHB	6.99/7.05	7.00/7.09	3.80/3.58	5.18/5.14
TSB	7.21/7.33	7.23/7.32	4.73/4.16	5.48/5.10
BHI	7.17/7.26	7.15/7.23	4.92/4.22	5.60/5.17
Minimal media				
MOPS	7.48	7.46	6.15	6.60
M9	7.27	7.23	6.53	6.61
M63	7.14	7.02	6.66	6.62
Davis	7.24	7.05	6.19	6.00

## Data Availability

The datasets generated during and/or analysed during the current study are available from the corresponding author on reasonable request.

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
