# Peer review of "Culture Media Composition Influences the Antibacterial Effect of Silver, Cupric, and Zinc Ions against Pseudomonas aeruginosa"

_biomolecules, 2022, doi:10.3390/biom12070963_

Round 1

Reviewer 1 Report

The manuscript is well written and organized. This is an interesting set of experiments to explore the authors evaluating the appropriability of culture media in testing the antibacterial activity of Ag+, Cu2+, and Zn2+ ions against Pseudomonas aeruginosa. They were able to identify that the most favorable medium for metal ions activity testing 20 was Luria-Bertani complex medium and MOPS minimal medium.

However, a few points are not clear to me, and would appreciate it if authors could clarify them more clearly in the manuscript:

My main issue is that some aspects could be further discussed, but I understand if the authors do not want to hypothesize without being able to validate such arguments.

Specific comments:

1.     There are many reports of low pH showing antimicrobial activity. In Figure 2, the pH-lowering effect of Cu2+ in LB is particularly strong, with a pH of less than 5 at the MIC250 shown in Table 1. In Fig. 3, it may be necessary to verify the effect of Cu2+ in pH 4 or lower conditions.

2.     I believe the authors should include statistical analysis even in the results and include the significant differences in Figure 3. And add more information in legend. For example, how many experiments have you done as described in materials and methods.

3.     In my opinion, as the author also states in the discussion part, measuring the concentration of metal ions in the medium may not be meaningful. Still, measuring the concentration of free cations produced from the added metal solution may be significant. The formation of insoluble compounds such as AgCl is also possible.

Author Response

Dear Editor,

We would like to express our sincerest gratitude to the reviewers for their thoughtful evaluation of our manuscript. We have considered all raised questions, and carefully revised the manuscript according to the reviewers’ comments.  Here follows the revised manuscript and detailed responses to these comments. Moreover, all changes we have made to the original manuscript are marked with red colour in the text.

Reviewer #1

Question 1. There are many reports of low pH showing antimicrobial activity. In Figure 2, the pH-lowering effect of Cu2+ in LB is particularly strong, with a pH of less than 5 at the MIC250 shown in Table 1. In Fig. 3, it may be necessary to verify the effect of Cu2+ in pH 4 or lower conditions.

Answer: According to reviewer’s suggestion, the pH of medium containing 250 µg/ml of Cu2+ was evaluated and the average value was equal to 5.083.  Moreover, it was found some sources (Adv. Environ. Biol., 8(13), 673-680, 2014; : DOI: 10.15666/aeer/1703_60816093; DOI: 10.12988/asb.2013.31043) showing that P. aeruginosa was able to growth even at pH 4.0 medium.

Q2. I believe the authors should include statistical analysis even in the results and include the significant differences in Figure 3. And add more information in legend. For example, how many experiments have you done as described in materials and methods.

Answer: Thank you for the suggestion. It was calculated the statistically significant differences using one-way-ANOVA and the Levene test and followed by the Tukey test in the OriginPro 2019b (OriginLab Corporation) software. The experiments were performed in triplicate and the p value was set at 0.05. The complete statistical analysis is presented in the Table S2 in the Supplementary Materials. Moreover, it was added information about number of experiments below the Figure 3 in the main manuscript and the description of the statistical test in Materials and Methods section.

Q3. In my opinion, as the author also states in the discussion part, measuring the concentration of metal ions in the medium may not be meaningful. Still, measuring the concentration of free cations produced from the added metal solution may be significant. The formation of insoluble compounds such as AgCl is also possible.

This is a very valuable comment. It has discussed the matter previously and agreed that the complete analysis should contain the measurement of the free metal cations concentration by ion-selective electrode, the measurement of total metal amount by ICP or ASA methods as well as the chemical analyses of precipitates. These values, put together, would make it possible to answer the question “What is happening in each medium?”.  Nevertheless, our goal was not to determine the absolute concentration of free ions in a given medium, but a relative change in their amount in different media, which affects the antimicrobial activity which can be observed as alterations in MICs’ values. Therefore, this paper focuses on microbiological analyses as such were goal of our study.

Reviewer 2 Report

review on the manuscript # biomolecules-1785436,entitled "Culture media composition influences the antibacterial effect of silver, cupric, and zinc ions against Pseudomonas aeruginosa" by J. Rewak-Soroczynska.   An interesting paper concerning the effects of growth media on antibacterial activity of metal cation. The results of this study shall contribute to achieving standardization in test methods for measuring antibacterial activity of metal cation.   Some comments on the manuscript   1) As the compositions of the media used have important information on discussion, Figs S1 and S2, as well Table S1 should be included in the main parts of the manuscript, not supplements.   2) in the section 3.2, l. 230~, " the theoretical analysis of ..." 2-1) Please describe roughly about the theory used in the theoretical analysis, or site references, otherwise it is impossible for readers to understand how the contents of the table 3 was derived. 2-2) In addition to the theoretical analysis, experimental analysis, such as X-ray Diffraction analysis and X-ray Photoelectron Spectroscopy, can detect the precipitates. Experimental results, if possible, to carry out, are more direct than theoretical investigations.   3) in the section 3.4 Experimental results obtained from the experiments only on Ag+ and Cu2+ were generalized for monovalent ion and divalent ion. Then authors should show experimental results for another monovalent ion as well as divalent ion, such as Zn2+ which was used in other parts of the study, or cite references which support your discussion.

4) l. 395~400

Inductively coupled plasma analysis nor AAS was mentioned in the ref. 20. correct references if necessary.

Recommendation: revision necessary.

Author Response

Dear Editor,

We would like to express our sincerest gratitude to the reviewers for their thoughtful evaluation of our manuscript. We have considered all raised questions, and carefully revised the manuscript according to the reviewers’ comments.  Here follows the revised manuscript and detailed responses to these comments. Moreover, all changes we have made to the original manuscript are marked with red colour in the text.

Reviewer #2

Question 1. As the compositions of the media used have important information on discussion, Figs S1 and S2, as well Table S1 should be included in the main parts of the manuscript, not supplements.   

Answer: Thank you for the suggestion. The figures were included in the main text (please see original manuscript). However, it was decided to move them to the Supplement in order to make the Manuscript more clear and easier to read. Nevertheless, following the Reviewer’s advice, it was decided to move mentioned figures back to the main text. The data visible in the Table S1 correspond with Figure S2, hence, we decided to leave the table in a current place (Supplement).

Q2.  In the section 3.2, l. 230~, " the theoretical analysis of ..." 2-1) Please describe roughly about the theory used in the theoretical analysis, or site references, otherwise it is impossible for readers to understand how the contents of the table 3 was derived. 2-2) In addition to the theoretical analysis, experimental analysis, such as X-ray Diffraction analysis and X-ray Photoelectron Spectroscopy, can detect the precipitates. Experimental results, if possible, to carry out, are more direct than theoretical investigations.

Answer: It should be mentioned here that before theoretical analysis there were performed numerous experiments where was observed what has been happening after mixing particular media components with metal cation and it was decided to add our observations to the Supplement (Table S2).  It was conducted the analysis basing on the observations (the chemical reactions) not only those visible as precipitation or color change but also that related to chemical interactions and they are listed in Table 3 in the main manuscript. Our goal was not only to find out what specific chemical compounds are in the precipitates but also to make a comparison between particular media and to identify the most probable media components that may be responsible for lowering the antibacterial impact of metal ions.

Q3.  in the section 3.4 Experimental results obtained from the experiments only on Ag+ and Cu2+ were generalized for monovalent ion and divalent ion. Then authors should show experimental results for another monovalent ion as well as divalent ion, such as Zn2+ which was used in other parts of the study, or cite references which support your discussion.

Answer: This is a very valuable comment. It has been soften and modified  the statement in the text as follows: “According to available report [32] it might be assumed that divalent ions could be much more prone to changes in antibacterial activity than monovalent ones.”

As was mentioned in the text : The Zn2+ was excluded from the analysis due to undetermined MIC values in all tested media (>1000 µg/ml)”. Firstly, it was included the zinc ions but in that case the analysis was incomplete and it was decided to exclude it.

Q4.  l. 395~400

Inductively coupled plasma analysis nor AAS was mentioned in the ref. 20. correct references if necessary.

Answer: In fact, the researchers (Hasman et al.) did not describe any of these methods (ICP OES or AAS) but used ion-selective electrode to measure the exact concentration of free copper ions in the medium and then they compared these values with the concentrations in the medium adding a certain amount of CuSO4 compound. This part of the Discussion has been rewritten and now it should be less misleading.

Round 2

Reviewer 1 Report

No further outstanding comments. The correct corrections and comments to the question have been made. This paper is considered suitable for publication in Biomolecules.